# Hierarchical Bandits for Adversarial Online Configuration Optimization

## Abstract

Motivated by dynamic parameter optimization in finite, but large action (configurations) spaces, this work studies the nonstochastic multi-armed bandit (MAB) problem in metric action spaces with oblivious Lipschitz adversaries. We propose *ABoB*, a hierarchical Adversarial Bandit over Bandits algorithm that clusters similar configurations to "virtual arms". In turn, it uses state-of-the-art existing "flat" MAB algorithms in each hierarchy to exploit local structures and adapt to changing environments. We prove that in the worst-case scenario, such clustering approach cannot hurt too much and ABoB guarantees a standard worst-case regret bound of $\mathcal{O}(k^{\frac{1}{2}}T^{\frac{1}{2}})$, where $T$ is the number of rounds and $k$ is the number of arms, matching the traditional flat approach. However, under favorable conditions related to the algorithm properties, clusters properties, and certain Lipschitz conditions, the regret bound can be improved to $\mathcal{O}(k^{\frac{1}{4}}T^{\frac{1}{2}})$. Simulations and experiments on a real storage system demonstrate that ABoB, can be made practical using standard algorithms like EXP3 and Tsallis-INF. ABoB achieves lower regret and faster convergence than the flat method, up to 50% improvement in known previous setups, nonstochastic and stochastic, as well as in our settings.

## 1 Introduction

The multi-armed bandit (MAB) problem is a fundamental concept in decision theory and machine learning. It effectively illustrates the exploration-exploitation dilemma that arises in many real-world situations, such as clinical trials, online advertising, resource allocation, and dynamic pricing Villar et al. (2015); Schwartz et al. (2017). In its simplest form, an agent must repeatedly choose from a set of actions, known as "arms," each of which provides a reward. The objective is to maximize the cumulative reward over time, or equally, minimizing the *regret*. The primary challenge is finding the right balance between exploring different arms to understand their reward distributions and exploiting those arms that have previously generated the highest rewards Slivkins et al. (2019); Bergemann & Valimaki (2006); Bubeck et al. (2012). The MAB framework has made significant theoretical progress over the years, and its practical applications offer valuable insights into optimal decision-making under uncertainty Djallel & Irina (2019). However, many real-world applications diverge from the assumptions of the classical MAB setting, especially when addressing evolving environments and complex relationships among choices Auer et al. (2002a); Slivkins & Upfal (2008).

This paper addresses one such departure, studying a framework for the multi-armed bandit problem in a dynamic, nonstochastic (adversarial) environment. The setup is inspired by real-world systems that involve a vast number of optimization parameters, such as automated configuration tuning for computing and storage systems. Automatic configuration and tuning in various real-world settings—ranging from industrial machines to smart home appliances—have become critical challenges due to the increasing complexity and diversity of modern systems and devices. In the context of data centers and high-performance computing (HPC), we can find examples of this, such as GPU kernel optimization using Bayesian optimization Willemsen et al. (2021), online energy optimization in GPUs Xu et al. (2024), and hyperparameter optimization Li et al. (2018). The method and work we report here were specifically applied to design an automated method that optimizes a real storage system employed in a large distributed computation cluster that has a large configuration space.

In our scenario, each system configuration (a specific setup of parameters) is considered as an arm. Configurations are situated in a metric space and partitioned into clusters that exhibit Lipschitz

(a) System overview  (b) Results' summary

Figure 1: (a) System overview of Adversarial Bandit over Bandits (ABoB). (left) Partition the configurations (arms) into clusters of similar configurations. (middle) Hierarchical bandits over the clusters (*virtual arms*). (right) Optimize a system's performance under a dynamic context. (b) A summary of ABoB's results.

properties, which allow us to handle the extensive search space. Unlike traditional bandit settings, our focus is on a nonstochastic landscape where the reward for each arm can change adversarially over time, including the optimal arm (configuration). This reflects, for example, the variation in performance of different system configurations under fluctuating workloads, such as the dynamic arrival and completion of job/tasks in the system. Importantly, the arms are not independent; they are grouped into clusters based on a predefined metric that represents the similarity of configurations. The arms' clustering can either be given as input or initially computed by the MAB algorithm.

In each cluster of arms, the rewards adhere to a Lipschitz condition. This means that configurations that are "close" (i.e., have similar parameter settings) will yield similar performance results, even as the performance shifts with the environment dynamics. This conveys that while the *optimal configuration* for a system may change with different workloads, configurations with similar parameter settings are likely to show comparable performance levels. The "traveling arms" framework requires algorithms that efficiently learn and adapt to an evolving reward landscape while taking advantage of the structural information offered by the partition and the Lipschitz characteristics of the arms. We explore the challenges and opportunities inherent in this new bandit setting, presenting algorithms and theoretical guarantees aimed at minimizing regret in this dynamic and practical environment.

To leverage the structural information inherent in the problem setup, we propose a hierarchical MAB algorithm called *Adversarial Bandit over Bandits* (ABoB). Refer to Figure 1 for an illustration. Our algorithm utilizes multiple instances of other well-known adversarial bandit algorithms like the classical EXP3 Auer et al. (2002a) or Tsallis-INF Zimmert & Seldin (2021), known to have the "best of two worlds" property. Given the arms' partition, at the first level (tier) of our hierarchy, a *parent* bandits algorithm (e.g., EXP3 or Tsallis-INF), looking at each cluster as a *virtual arm*, determines which cluster to engage in the current time step. Subsequently, a second-level algorithm, denoted as a *child* bandit algorithm (e.g., EXP3 or Tsallis-INF), specific to each cluster, is activated to select the next arm within the chosen cluster. By design, employing known adversarial algorithms allows us to benefit from their known strengths in adversarial settings. At the same time, the clustering approach enables us to exploit the underlying metric, as similar configurations tend to yield similar outcomes (i.e., they satisfy the Lipschitz condition). While the fundamental concept of our algorithm is easy to understand, we have not encountered similar solutions or setups in existing literature.

**Paper Contributions.** This paper presents several key contributions: (i) *A Lipschitz Clustering-based MAB Variant*: We introduce a new variant of the Multi-Armed Bandit (MAB) problem that addresses the challenges of dynamic environments with large, structured action spaces, particularly in the context of system configuration optimization. (ii) *Hierarchical Clustering and the ABoB Algorithm*: To tackle the aforementioned problem, we propose the ABoB algorithm, which is based on classic MAB algorithms, but additionally employs hierarchical clustering. We provide a general theoretical bound for ABoB, demonstrating that it can achieve a worst-case regret of $\mathcal{O}(\sqrt{Tk})$, where $T$ represents the number of iterations and $k$ denotes the number of arms that is finite but assumed to be large. This performance matches the bound of the "flat" method (e.g., of Tsallis-INF Zimmert & Seldin (2021) while offering significant potential for enhancements in structured environments. (iii) *Improved Bound for Lipschitz Clusters*: We further analyze ABoB's performance under assumptions on the properties of the MAB algorithms it uses, and the Lipschitz condition satisfied by the clusters. This analysis leads to an improved regret bound that can be up to $\mathcal{O}(\sqrt[4]{k})$

better than the general bound. This highlights the algorithm's capacity to leverage the metric space structure and Lipschitz properties for superior performance. Overall, when using ABoB, there's not much to lose in the worst case, but much to gain in special cases. (iv) *Empirical Validation*: We performed an extensive simulation study to validate our theoretical findings. In addition to well-known synthetic settings (both stochastic and nonstochastic), we present experimental results conducted on a real storage system tasked with optimizing access to remote physical disks. These experiments demonstrate the practical effectiveness of ABoB, showing its ability to achieve lower regret and faster convergence compared to traditional MAB algorithms in a real-world setting, with improvements of up to 91% in our scenarios. See Figure 1(b) for summary of results. Due to space constraints, proofs, extra figures, and additional experiments are present in the technical appendix.

## 2 RELATED WORK

The multi-armed bandit (MAB) problem has been extensively studied under various assumptions, leading to a rich landscape of algorithms and theoretical results Slivkins et al. (2019). Early work focused primarily on the stochastic setting, where the finite set of arm rewards are IID random variables drawn from fixed, unknown distributions. Algorithms like Upper Confidence Bound (UCB) Auer et al. (2002b) and Successive Elimination Even-Dar et al. (2002) provide strong performance guarantees in this scenario by balancing exploration and exploitation based on estimated reward distributions' means and confidence intervals. The canonical worst-case scenario of "needle in a haystack" leads to a regret of $\tilde{\mathcal{O}}(\sqrt{kT})$ (ignoring polylogarithmic factors) for these algorithms.

Another line of work deals with continuous action spaces, still under stochastic settings. The most straightforward approach is performing uniform discretization on the arms, under assumptions of metric spaces and smoothness assumptions, and then applying a known multi-armed bandit approach (like UCB). More advanced techniques include approaches like the *zooming algorithm* Kleinberg et al. (2008; 2019), which takes a more adaptive and dynamic approach. It starts with a coarse view of the action space and progressively refines its focus on promising regions. It maintains a confidence interval for each arm (or region), and based on these intervals, it either "zooms in" on a region by dividing it into subregions or eliminates it if it's deemed suboptimal. A more challenging research direction considers the finite nonstochastic setting with a finite number of arms, where adversarial opponents control reward assignments. For this setting to be feasible, the usual assumption is of an oblivious adversary and a *regret* where the algorithm is compared with the best arm in hindsight. For this case, algorithms like EXP3 Auer et al. (2002a) provide robust regret guarantees against any sequence of rewards, surprisingly matching the worst-case bound of the stochastic case.

More recently, and perhaps the most related approach to the algorithm presented in this paper, the combination of continuous action spaces and adversarial rewards has been explored in the adversarial zooming setting, presenting unique challenges addressed by algorithms that dynamically adapt their exploration strategy based on the observed structure of the reward landscape Podimata & Slivkins (2021), yielding a regret bound that depends on a new quantity, $z$, called *adversarial zooming dimension* and is given by $\mathbb{E}[R(T)] \leq \tilde{\mathcal{O}}(T^{\frac{z+1}{z+2}})$. For *finite* number of arms, we have $z = 0$, and the paper provides a worst-case regret-bound of $\mathcal{O}(\sqrt{kT \log^5 T})$. This regret bound is similar to that of the nonstochastic multi-armed bandit, i.e. $\tilde{\mathcal{O}}(\sqrt{kT})$. For the finite case, the algorithm, therefore, does not offer improved bounds. In contrast, our approach provides conditions for improved performance and demonstrates practical benefits.

Other papers studied various settings of correlated or dynamic arms, for example, $\mathcal{X}$–Armed Bandits Bubeck et al. (2011), Contextual Bandits Slivkins (2011), Correlated arms Gupta et al. (2021) Eluder Dimension Russo & Van Roy (2013) and Dependent Arms Pandey et al. (2007), to name a few, but none are using our exact setup, nor the novel idea of parent-child bandits.

## 3 PROBLEM FORMULATION AND THE ABOB ALGORITHM

We study the Adversarial Lipschitz MAB (AL-MAB) Problem Podimata & Slivkins (2021). In particular, we consider the finite case where the set of arms forms a metric space, and the adversary is oblivious to the algorithm's random choices. The problem instance is a triple $(K, \mathcal{D}, \mathcal{C})$, where $K$ it the set of $k$ arms $\{1, 2, \ldots, k\}$, and $(K, \mathcal{D})$ is a metric space and $\mathcal{C}$ is an expected rewards

assignment, i.e., an infinite sequence $\mathbf{c}_1, \mathbf{c}_2, ...$ of vectors $\mathbf{c}_t = (c_t(1), ..., c_t(k))$, where $c_t(a)$ is the expected reward of arm $a$ at time $t$, and $\mathbf{c}_t : K \to [0, 1]$ is a Lipschitz function on $(K, \mathcal{D})$ with Lipschitz constant $1$ at time $t$. Formally,

$$|c_t(a) - c_t(a')| \leq \mathcal{D}(a, a') \tag{1}$$

for all arms $a, a' \in K$ and all rounds $t$, where $\mathcal{D}(a, a')$ is the distance between $a$ and $a'$ in $\mathcal{D}$.

For our analytical results, we consider a special case where we partition the set of arms into a set of clusters. We let $\mathcal{P}$ be a partition[1] of $K$ where $p$ is the number of clusters and $P^1, \ldots, P^p$ are the mutually exclusive and collectively exhaustive clusters where for each $1 \leq i \leq p$, $P^i \subseteq K$ and $|P^i| > 0$. We will show results for the case of arbitrary clustering, as well as the case that the clustering forms a metrics space.

For such an adversarial setting, the standard metric of interest is the *regret* (aka weak regret in the original work of Auer et al. (2002a)), $R(T)$ defined for a time horizon $T$. To define it, we first need to determine the reward of an algorithm $A$. For an algorithm $A$ that selects arm $a_t$ at time $t$ we consider its reward as: $G_A(T) \overset{\text{def}}{=} \sum_{t=1}^{T} c_t(a_t)$. The best arm, in hindsight, is defined as $G_{\max}(T) \overset{\text{def}}{=} \max_{a \in K} \sum_{t=1}^{T} c_t(a)$, and, in turn, the *regret* of algorithm $A$ is defined as

$$R(T) \overset{\text{def}}{=} G_{\max}(T) - G_A(T). \tag{2}$$

We will be interested in the expected regret $G_{\max}(T) - \mathbb{E}[G_A(T)]$ Auer et al. (2002a).

Next, we present our novel algorithm, ABoB, and formally study its performance. The basic concept of the ABoB algorithm is based on a simple idea of *divide-and-conquer*. Alongside the set of arms, we receive or create a partition (e.g., by using your favorite clustering algorithm) of the arms into clusters. This partition creates, in fact, a hierarchy. The first level is the "clusters" level, where we can see each cluster as a virtual (adversarial) single arm based on the collective of arms in it. The second level is the level of physical arms within each cluster. See Algorithm 1 and Figure 1.

ABoB uses this hierarchy to run flat Adversarial MAB (A-MAB) algorithms for each level, denoted as *parent A-MAB* and *child A-MAB*, for the first and second level, respectively. The particular A-MAB algorithms used in ABoB can differ between levels, and the idea is to consider well-known A-MAB algorithms like EXP3 Auer et al. (2002a) and its variations Seldin & Lugosi (2017) or Tsallis-INF Zimmert & Seldin (2021) for example. For the simplicity of presentation and as a concrete example, we use the classical EXP3 Algorithm Auer et al. (2002a) for both levels, unless otherwise stated. For the first level, there is a single EXP3, the parent A-MAB algorithm, that on each time step, selects between the virtual arms generated by the clusters and decides from which cluster it will sample the next arm. In turn, for each cluster, there is a second-level EXP3 algorithm that selects the next arm within the cluster, but it is activated only when the first level selects that cluster. Upon selecting an arm, we update the arm's reward and then the child algorithm parameters, like arms weights in the EXP3 and Tsallis-INF algorithms (relative to the arms in the current cluster). Next, we update the reward of the virtual arm of the cluster and the parent algorithm's parameters (relative to other virtual arms (clusters)) and move to the next time step.

**Correctness and Time Complexity of ABoB.** The correctness of the algorithm follows from the observation that each cluster can be viewed as an adversarial virtual arm, that generates a reward when activated at time $t$. Therefore, since the parent A-MAB algorithm is adversarial, the algorithm setup is valid. We note that regarding the child A-MAB algorithm, we have more flexibility, and if we know, for example, that each cluster is stochastic, we can use algorithms that may better fit this environment, like UCB Auer et al. (2002b). An important feature of ABoB is its running time. In the adversarial setting the running time of many algorithms is $\mathcal{O}(kT)$, where $T$ updates, each in the order of $\mathcal{O}(k)$ are required (e.g., in EXP3). In ABoB, however, we still need $T$ updates, but each is of the order of $\mathcal{O}(\text{\# of clusters} + \text{\# of arms in the selected cluster})$, this for example can be significantly lower then $\mathcal{O}(k)$, e.g., when we use $\sqrt{k}$ clusters each of size $\sqrt{k}$ (up to rounding).

The main question we answer next is: In terms of the regret, how much do we lose or gain by using ABoB and the hierarchical approach? Somewhat surprisingly, we show in Section 4.1 that not much is lost in the worst case, and in Section 4.2 we prove that we can gain significantly under certain Lipschitz conditions and A-MAB algorithms.

---

[1]We use partition and clustering interchangeably.

---

**Algorithm 1** ABoB: Hierarchical Adversarial MAB Algorithm

---

**Input:** Parent A-MAB and Child A-MAB algorithms: Adversarial multi-armed bandits algorithms (e.g., EXP3, Tsallis-INF), $k$ arms and their partition, $\mathcal{P}$, into clusters
1: **Initialize**: Init the parent A-MAB algorithm (weights, etc.). For each cluster, initialize its own child A-MAB algorithm (weights, etc.)
2: **for** $t = 1$ to $T$ **do**
3:   (a) Cluster selection using the parent A-MAB algorithm on virtual arms (clusters). Let $P_t \in \mathcal{P}$ be the selected cluster.
4:   (b) Arm selection using the child A-MAB algorithm on physical arms in $P_t$. Let $a_t \in P_t$ be the selected arm.
5:   (c) Pull $a_t$, observe reward $c_t$.
6:   (d) Update the arms-level parameters (e.g., weights) for physical arms in $P_t$ using $c_t$.
7:   (e) Update cluster-level parameters (e.g., weights) for virtual arms (clusters) using $c_t$.

---

# 4 ANALYTICAL BOUNDS FOR THE ABoB ALGORITHM

In this section we study ABoB analytically. To continue, we need additional notations. Recall that $T$ is the time horizon, $\mathcal{P}$ is a partition of the set of arms $K$, $k$ in the number of arms, and $p$ is the number of clusters. Let $T^i$ be the ordered set of times by which the algorithm visits cluster $i$, so $\sum |T^i| = T$. We denote by $t(i,j) \in T^i$, the time in which the algorithm visited cluster $i$ for the $j$th time. We then additionally define $a^* \in P^*$ as the best arm in hindsight, i.e., the arm that maximizes Eq. equation 2, and $P^*$ as the cluster that contains $a^*$.[2] Let $p^* = |P^*|$ denote the size of the cluster that holds $a^*$. Let $G_{\mathrm{maxP}}(T)$ be the expected reward of the best cluster in hindsight:

$$G_{\mathrm{maxP}}(T) \stackrel{\text{def}}{=} \max_{i \in K} \sum_{t=1}^{T} c_t(P^i) = \sum_{t=1}^{T} c_t(P^+),$$

and $P^+$ be the best cluster in hindsight, the cluster (virtual arm) that should have been played constantly by the first-level, parent A-MAB algorithm (e.g., EXP3), assuming it is played internally by its own second-level, child A-MAB algorithm (e.g. EXP3).

Lastly, for each cluster $P^i$, we define its best expected reward in hindsight $G_{\mathrm{maxP}^i}$; what was its expected reward if we were playing its best arm in hindsight (at the times we visited it), formally,

$$G_{\mathrm{maxP}^i}(T^i) \stackrel{\text{def}}{=} \max_{a \in P^i} \sum_{j=1}^{|T^i|} c_{t(i,j)}(a).$$

In a hierarchical multi-armed bandit setup of the ABoB algorithm, the total regret can be expressed as the sum of two components: (i) Regret for choosing a cluster other than the best one: This is the regret incurred by the first-level bandit in not selecting the best cluster, defined as the cluster containing the best arm in hindsight. (ii) Regret within the cluster: This is the regret incurred by the second-level bandit in selecting an arm within the chosen cluster relative to the best arm in that cluster. Next, we use this observation to bound the regret of ABoB in different scenarios.

## 4.1 ARBITRARY CLUSTERING: NOT MUCH TO LOSE

In this subsection, we study the regret where the partition (or clustering) is done arbitrarily, i.e., without assuming any metric between the arms. For EXP3, the analysis shows (Theorem 4.2) that even in the case of arbitrary partition of the arms, the asymptotic regret is equivalent in the worst-case to those achieved by the "flat" EXP3 algorithm, i.e., $\mathcal{O}\left(\sqrt{kT \log k}\right)$ Auer et al. (2002a).

The expected regret of ABoB can be bounded as follows: For each level of the hierarchy, we analyze its regret resulting from the relevant A-MAB algorithms. At the first level, the regret is of the parent A-MAB playing with $p$ virtual arms of the clusters, and with respect to $G_{\mathrm{max}}$. At the second level, we compute the regret within each cluster $P^i$ using its child A-MAB and with respect to the best cost of the cluster in hindsight $G_{\mathrm{maxP}^i}$. Formally,

---

[2]If $a^*$ is not unique, we consider $P^*$ to be the largest cluster that contains an $a^*$.

- *First level*: the regret of not choosing the best cluster in hindsight $P^+$ during all times up to $T$:

$$R_1^a \overset{\text{def}}{=} G_{\text{maxP}}(T) - \sum_{t=1}^{T} c_t(P_t) = G_{\text{maxP}}(T) - \sum_{i=1}^{p} \sum_{j=1}^{|T^i|} c_{t(i,j)}(P^i),$$

plus the regret for not choosing the cluster $P^*$ and selecting within always arm $a^*$:

$$R_1^b \overset{\text{def}}{=} G_{\text{max}}(T) - G_{\text{maxP}}(T)$$

- *Second level*: the regret in each cluster not playing its best arm in hindsight all the time:

$$R_2 \overset{\text{def}}{=} \sum_{i=1}^{p} \left( G_{\text{maxP}^i}(T^i) - \sum_{j=1}^{|T^i|} c_{t(i,j)}(a_{t(i,j)}) \right)$$

Formally, we can claim the following.

**Claim 4.1.** *For the ABoB algorithm that uses EXP3, the following holds:* (1) $\mathbb{E}\left[R_1^a\right] \leq \mathcal{O}\left(\sqrt{pT \log p}\right)$. (2) $\mathbb{E}\left[R_1^b\right] \leq \mathcal{O}\left(\sqrt{p^*T \log p^*}\right)$. (3) $\mathbb{E}\left[R_2\right] \leq \mathcal{O}\left(\sqrt{kT \log(k/p)}\right)$.

Based on Claim 4.1, we can bound the regret of ABoB.

**Theorem 4.2.** *For $k$ nonstochastic arms, $T > 0$ and a partition $\mathcal{P}$ of the arms, the regret of the ABoB algorithm using EXP3 is bounded as follows:*

$$G_{\text{max}} - \mathbb{E}[G_{\text{ABoB}}] \leq \mathcal{O}\left(\sqrt{pT \log p}\right) + \mathcal{O}\left(\sqrt{p^*T \log p^*}\right) + \mathcal{O}\left(\sqrt{kT \log(k/p)}\right)$$

*where $p$ is the number of clusters and $p^*$ is the size of the cluster with the best arm in hindsight.*

Since $p, p^* \leq k$, the worst case bound is in the same order as the flat case. Moreover, we will usually use $p, p^* \ll k$, and the contribution of the first two terms should be smaller than the third. With this, we can state the first main takeaway of our approach.

■ **Takeaway.** *Using cluster does not "hurt" much the overall regret of the* flat *approach.*

We note that similar results can be obtained for other A-MAB algorithms like Tsallis-INF Zimmert & Seldin (2021), removing the logarithmic factors. Still, the question remains if we can benefit from clustering; the following subsection answers this question affirmatively.

### 4.2 CLUSTERING WITH LIPSCHITZ: MUCH TO GAIN

In this subsection, we consider the case that the rewards of the arms form a metric space and, in particular, the simpler case where, within each cluster, we have a *Lipschitz* condition ensuring that the rewards of all arms in a cluster are "close" to each other. For this simpler case, we do not even require a condition for the distance between clusters. Such a situation fits, for example, in a system's parameters optimization problem where each arm corresponds to a set of parameters trying to optimize a certain objective (e.g., power, delay, etc.) (See Section 6). The set of potential arms is extensive since we can tune each parameter to many possible values. In turn, we assume a smoothness condition on each parameter, such that each minor change of a parameter results in a minor shift in the outcome. The "state" of the system (e.g., number of active jobs) is unknown and can change over time (e.g., as new jobs come and go), even dramatically, but for every state, the Lipschitz condition holds (Eq. equation 1). More formally, for the clustering scenario, we say that an arms partition $\mathcal{P}$ is an $\ell$-partition if the following Lipschitz condition holds:

$$\forall P \in \mathcal{P}, a, b \in P \quad |c_t(a) - c_t(b)| \leq \ell, \quad \forall \text{ rounds } t. \tag{3}$$

Following Theorem 4.2, and since within each cluster, the expected regret is at most $\ell$, we can easily state the following.

**Corollary 4.3.** *For $k$ nonstochastic arms, $T > 0$ and an $\ell$-partition $\mathcal{P}$ of the arms, the regret of the ABoB algorithm using EXP3 is bounded as follows:*

$$G_{\text{max}} - \mathbb{E}[G_{\text{ABoB}}] \leq \mathcal{O}\left(\sqrt{pT \log p}\right) + \mathcal{O}\left(\sqrt{p^*T \log p^*}\right) + T \cdot \ell$$

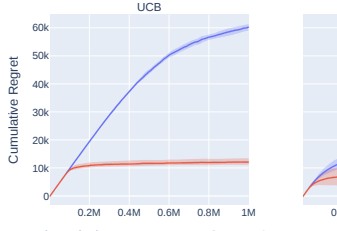 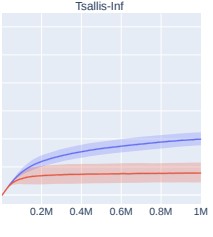 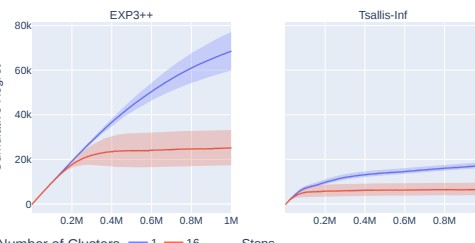

Figure 2: Comparing ABoB (with $\sqrt{k} = 16$ clusters) with "flat" algorithms (single cluster) for UCB and Tsallis-INF algorithms. Stochastic Scenario, $k = 256$.

Figure 3: Fixed Optimal Arm, and Nonstochastic Scenario, $k = 256$. Comparing ABoB (with $\sqrt{k} = 16$ clusters) with "flat" algorithms for EXP3++ and Tsallis-INF algorithms.

Note that the above bound does not assume any contribution to the execution of EXP3 within clusters, resulting from the $\ell$-partition. Using an appropriate A-MAB algorithm within each Lipschitz cluster potentially allows us to improve the result. Let ALB$^+$ denote an adaptive Lipschitz bandits algorithm with the following property:

*Property* 1. For the Adversarial Lipschitz MAB setting with a Lipschitz condition $\ell$ (Eq. 1) the algorithm regret is bounded by $\mathcal{O}\left(\ell \cdot \sqrt{kT \log k}\right)$.

Note that for a flat ALB$^+$ algorithm (a single cluster) $\ell$ might be large (e.g., 1), but when we break the arms into clusters, within each cluster $\ell$ might be significantly smaller. The following theorem formalized the potential gains.

**Theorem 4.4.** *For $k$ nonstochastic arms, $T > 0$ and an $\ell$-partition $\mathcal{P}$ of the arms, the regret of the ABoB, using an ALB$^+$ algorithm can be bounded as follows:*

$$G_{\max} - \mathbb{E}[G_{\text{ABoB}}] \le \mathcal{O}\left(\sqrt{pT \log p}\right) + \mathcal{O}\left(\ell\sqrt{p^*T \log p^*}\right) + \mathcal{O}\left(\ell \cdot \sqrt{kT \log(k/p)}\right), \quad (4)$$

*where $p$ is the number of clusters and $p^*$ is the size of the cluster with the best arm in hindsight.*

Following Theorem 4.4 we can state the second main takeaway of the paper.

■ **Takeaway.** *The results hold for nonstochastic arms, which keeps the Lipschitz condition. Comparing the upper bound of the "flat" (single cluster) ALB$^+$ algorithm, we can achieve improvement when the number of clusters and the size of the "best" cluster are relatively small. i.e., $p \ll k$, and $p^* \ll k$, and when the Lipschitz constant is sufficiently small. i.e., $\ell \le \frac{1}{\sqrt{k}}$.*

As a concrete example, we can state the following

**Corollary 4.5.** *Given an $\ell$-partition $\mathcal{P}$, with $\ell \le \frac{1}{\sqrt[4]{k}}$ and $\sqrt{k}$ clusters, each of size $\sqrt{k}$, the ABoB, using an ALB$^+$ algorithm can be bounded as $\mathcal{O}(\sqrt{k^{1/2}T \log k})$, which is $\Omega(\sqrt[4]{k})$ time better than the flat ALB$^+$ algorithm.*

In the next sections, we turn to simulation and experimental results to validate the performance of ABoB in concrete examples. A subtle point is what ALB algorithm to use within each cluster. We are not aware of an algorithm that formally fulfills Property 1 for the Lipschitz condition. Therefore, we choose to use EXP3 as it satisfies a weaker Lipschitz condition where an a priori translation and rescaling are needed Auer et al. (2002a) is required to reduce the regret. Nevertheless, as shown in the next section, clustering the arms and using EXP3-based or Tsallis-INF for parent and child clusters can still significantly improve the algorithm's performance.

## 5 EMPIRICAL STUDY

In this section (and the appendix), we report on an extensive empirical study on the performance of ABoB compared to flat algorithms. We consider syntactic scenarios in increasing complexity, and in Section 6, we present results based on real system measurements. We report results on three

adversarial algorithms Tsallis-INF Zimmert & Seldin (2021), EXP3 Auer et al. (2002a), and EXP3++ Seldin & Lugosi (2017), but also on the, well-known, UCB1 Auer et al. (2002b) algorithm for the stochastic case. In turn, we consider different ABoB algorithms, including the above MAB algorithms. We mostly concentrate on the case where both the parent and child algorithms are the same, and in particular on the Tsallis-INF algorithm, which is the state-of-the-art algorithm enjoying the best-of-both-worlds regret bounds (i.e., it is optimal both for stochastic and adversarial settings). Unless otherwise stated, we consider Bernoulli r.w. arms with different settings on their mean values. The default number of arms is $k = 256$, and the default number of steps is $T = 10^6$. We repeated each experiment 10 times and report the average and standard deviation. Additional and more extensive figures, including non-identical parent-child algorithms, are given in Appendix A.3.

**Stochastic Scenario.** The first experiment, shown in Figures 2 (and Figure 6 in Appendix) is a standard stochastic MAB setting Zimmert & Seldin (2021), where the mean rewards are $\frac{1+\Delta}{2}$ for the single optimal arm and $\frac{1-\Delta}{2}$ for all the suboptimal arms, where $\Delta = 0.1$. Other values for $\Delta$ and $k$ are reported in the appendix. We can clearly observe the performance improvement of ABoB in all Algorithms for $p = \sqrt{k} = 16$ clusters each of size $\sqrt{k} = 16$. As was reported, for a single cluster, Tsallis-INF produces the best results as in Zimmert & Seldin (2021), but ABoB using Tsallis-INF with $\sqrt{k} = 16$ further improves the regret.

**Fixed Optimal Arm and Nonstochastic (adversarial) Scenario.** The second experiment, also taken from Zimmert & Seldin (2021), considers a non-stochastic (adversarial) environment with a single fixed optimal arm but changing mean values. The mean reward of (optimal arm, all sub-optimal arms) switches between $(\Delta, 0)$ and $(1, 1 - \Delta)$, while staying unchanged for phases that are increasing exponentially in length. We set $\Delta = 0.1$ and other values for $\Delta$ and $k$ are reported in the appendix. Figure 3 (and Figure 7 in the Appendix) presents the results for several MAB algorithms, where Tsallis-INF again achieves the best results as in Zimmert & Seldin (2021), and ABoB improves it.

**"Traveling" Optimal Arm, Nonstochastic (adversarial) and Metric Spaces.** In the third experiment, we consider a scenario motivated by our application for configuration tuning. We consider a Nonstochastic (adversarial) reward with changing optimal arms (i.e., "traveling arms"), but all rewards are in a metric space and, in particular, follow a Lipschitz condition. More formally, we have used the following setup for the environment: (i) Metric Space: The metric space is defined as a hypercube of dimension $d$ $\mathcal{Q} = [0, \frac{1}{\sqrt{d}}]^d$. We use a setup such that $\sqrt[d]{k}$ is an integer. (ii) Arms' location: The $k$ arms are placed over an equally spaced grid such that the distance between two closest points in each dimension is $w = \frac{1}{\sqrt{d}k^d}$. For arm $i$, we denote its location on the grid as $x_i$ and recall that its mean reward at time $t$ is denoted as $c_t(x_i)$ (or $c_t(i)$). (iii) Arm's reward distribution: We define the mean reward to be $c_t(x_i) = 1 - \|a^*(t) - x_i\|$, where $a^*(t)$ is a point in the (continuous) cube that represents the best $d$-dimensional configuration settings at time $t$. Note that $a^*(t)$ is a parameter of the optimization problem and could be *dynamic* over time. (iv) Non-Stochastic: to simulate a dynamic system, we have changed $a^*(t)$ over time using a normal random walk, such that $a^*(t + 1) = a^*(t) + n(t)$ and $n(t)$ is a $d$-dimensional normally distributed random variable with zero mean and equal diagonal covariance $\sigma^2$ (keeping $a^*(t) \in \mathcal{Q}$ by clipping).

Unless otherwise stated, we clustered the $d$-dimensional cube into equal volume sub-spaces. For ABoB we evaluated an increasing number of clusters $p$, from $p = 1$ to $p = k$, note that for these extreme cases it coincides with the flat A-MAB algorithm. The default setting we used was $k = 256$ arms, and the clusters varied as $p = 2^j$, where $j \in \{0, 1, \ldots, \log_2 k\}$. In order to further validate the theoretical results on a nonstochastic, adversarial setup, we have changed the system parameter $a^*(t)$ over time such that the different arms' mean reward changes over time, and thus also the best arm. In the appendix, Figures 8(b) and 8(a) depicts this by showing the arms' mean reward over time as well as the optimal arm index that changes over time and the location of the best arm in the parameter space and how often an arm was the best. Figure 4(a) presents the results of the above setup comparing the regret of ABoB with Tsallis-INF as a function of number of steps. Additionally, Figure 4(b) shows the regret as a function of the number of clusters, which shows that using 16 clusters, ABoB achieves a regret of $435 \pm 69$ compared to the flat Tsallis-INF baseline which achieves a regret of $4879 \pm 259$, again showing the potential gain (t-test p-value $8.2 \times 10^{-14}$), of about 91%. Notice that the regret can be negative since we compare the results of Tsallis-INF to the best *fixed* arm while the optimal arm is traveling. Since Tsallis-INF can "travel" as well, its result can be better than the fixed optimal.

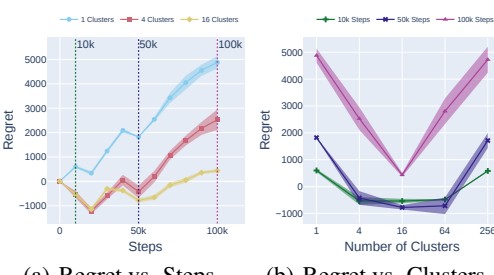
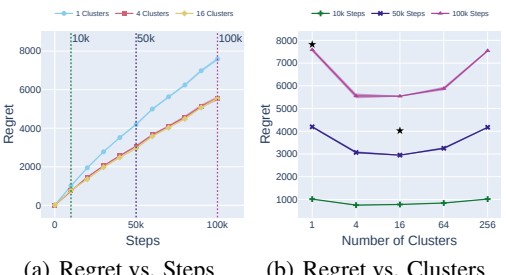

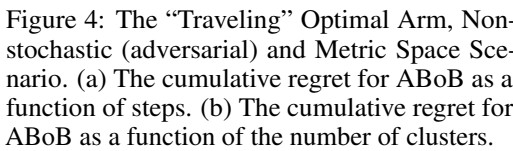

(a) Regret vs. Steps  (b) Regret vs. Clusters  (a) Regret vs. Steps  (b) Regret vs. Clusters

Figure 4: The "Traveling" Optimal Arm, Non-stochastic (adversarial) and Metric Space Scenario. (a) The cumulative regret for ABoB as a function of steps. (b) The cumulative regret for ABoB as a function of the number of clusters.

Figure 5: Results from a real storage system. (a) The cumulative regret of ABoB for reward from a real system. (a) The regret as a function of the number of clusters. Star-shaped points are the regret results of runs on the real system.

# 6    EXPERIMENTAL RESULTS: A REAL SYSTEM

We compare Tsallis-INF and ABoB using Tsallis-INF on a real storage system with the goal of maximizing the performance of the system. The system has a large configuration space of several parameters, some continuous and some discrete (integer value), where each performance evaluation takes a few seconds. Furthermore, the system undergoes dynamically changing workloads, which affect the observed reward. Here we report the results of optimizing 2 parameters while keeping the other fixed. Each run consists of $100,000$ iterations with 256 arms generated uniformly at random from the system's configuration search space. Clustering was implemented using K-means over the normalized arms parameters. The system cycles through six distinct workloads, switching approximately every 10,000 iterations.

**Rewards in a Real System.** In order to validate that the real system follows the Lipschitz condition, we estimate the Lipschitz's constant $\ell$ in the real arm reward mean and compare it to a shuffled reward distribution. To estimate $\ell$, we iterated over the arms, and for each arm $x_i$ over its $n$ nearest neighbors, $x_j \in \mathcal{N}_n(i)$. For each arm, we then computed the mean ratio between the rewards and the metric: $\ell_i = \frac{1}{n}\sum_{j\in\mathcal{N}_n(i)}(|r_j - r_i|)/(|x_j - x_i|)$. In the Appendix, Figure 9(a) shows the distribution of $\ell_i$ for both the real rewards shown in Figure 9(b) and a random permutation of the rewards. The figure validates that the Lipschitz assumptions hold for the real system as the range of $\ell_i$ is small.

**Flat Tsallis-INF vs. ABoB with Tsallis-INF: Real system data.** Using the setup described above (for $T = 100K$ and the number of arms is $k = 256$), we run the algorithms on the *real system*. The star-shaped points in Figure 5(b) shows that Tsallis-INF obtained a regret of 7819, while ABoB, using 16 clusters, provided a regret of 4025. This is an improvement of about 49%. The regret was computed by comparing the algorithms' rewards to the best empirical arm in hindsight, where the rewards of each arm are interpolated between measured samples. To validate the approach further, we replayed the reward sequence that was recorded from the real system (again using interpolation to fill rewards in all time steps). Figures 5(a) and 5(b) shows how the ABoB ($5543 \pm 21$) again is dominant over the flat Tsallis-INF baseline ($7584 \pm 79$) (t-test p-value $1.8 \times 1.3^{-15}$), about 27% improvement.

# 7    CONCLUSION

This paper introduced a novel nonstochastic, metric-based MAB framework to address the challenge of optimizing decisions in dynamic environments with large, structured action spaces, such as automated system configuration. We proposed ABoB, a hierarchical algorithm that leverages the clustered, Lipschitz nature of the action space. Our theoretical analysis demonstrated that ABoB achieves robust worst-case performance, matching traditional methods, while offering significant improvements when the underlying structure is favorable, as demonstrated by an improved regret bound under Lipschitz conditions. Importantly, these theoretical findings were validated through both simulations and experiments on a real storage system, confirming ABoB's ability to achieve lower regret and faster convergence in practice. Future work includes exploring adaptive clustering techniques, multilevel hierarchies of clusters, and studying distributed settings.

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

# A  TECHNICAL APPENDIX

## A.1  PROOFS

*Proof of Claim 4.1.* The bound in equation 1 follows from running EXP3 on the clusters, each as a virtual arm. There are $p$ clusters, and the regret is with respect to playing the best cluster (virtual arm), $P^+$ in hindsight, so the results directly follow from the EXP3 bound. For the bound in equation 2, we note that,

$$\mathbb{E}[G_{\max} - G_{\max P}(T)] = G_{\max} - \mathbb{E}[\sum_{t=1}^{T} c_t(P^+)] \leq$$

$$\leq G_{\max} - \mathbb{E}[\sum_{t=1}^{T} c_t(P^*)] \leq \mathcal{O}\left(\sqrt{p^* T \log p^*}\right),$$

where the first inequality follows since cluster $P^+$ has the largest expected reward (by definition), at least as large as cluster $P^*$, and the last inequality follows from the EXP3 bound in the cluster $P^*$ for which $a^* \in P^*$. For the equation in equation 3, the result follows from running EXP3 in each cluster

$$\mathbb{E}\left[\sum_{i=1}^{p}\left(G_{\max P^i}(T^i) - \sum_{j=1}^{p} \sum_{t'=t(i,j)} c_{t'}(a_{t'})\right)\right] \leq$$

$$\leq \sum_{i=1}^{p} \mathcal{O}\left(\sqrt{|P^i||T^i|\log|P^i|}\right) \leq \mathcal{O}\left(\sqrt{kT\log\frac{k}{p}}\right), \tag{5}$$

where the first inequality is the EXP3 bound, and the last inequality follows from the concavity of the function and under the constraints $\sum|\mathcal{P}^i| = k$ and $\sum|T^i| = T$, the worst case is $\forall i, |\mathcal{P}^i| = \frac{k}{p}$ and $|T^i| = \frac{T}{p}$. □

*Proof of Theorem 4.4.* The proof follows from Theorem 4.2, the proof of Eq. 5, and Property 1. We can improve Eq. 5 by plugging Property 1 so for each cluster $i$ its internal regrate is bounded by $\mathcal{O}\left(\ell \cdot \sqrt{|\mathcal{P}^i||T^i|\log|\mathcal{P}^i|}\right)$. In turn, the worst case for the sum $\sum_{i=1}^{p} \mathcal{O}\left(\ell\sqrt{|P^i||T^i|\log|P^i|}\right)$ is still the case where $\forall i, |\mathcal{P}^i| = \frac{k}{p}$ and $|T^i| = \frac{T}{p}$ and the results follows. □

## A.2  COMPUTING INFRASTRUCTURE USED FOR RUNNING EXPERIMENTS

All experiments were run on a standard MacBook Pro laptop with Apple M2 Pro Chip, 12 Cores, and 32 GB Memory.

## A.3  MORE DETAILS FOR FIGURES WITHIN THE PAPER

Figure 6 extends Figure 2, showing the behavior of the EXP and EXP++ algorithms. Figure 7 extends Figure 3, showing the behavior of the EXP and UCB algorithms.

Figure 8 provides an example of the "traveling arm" scenario where the best arm changes in time.

Figure 9 provides additional details on the dataset and the results from a real storage system.

## A.4  ONE DIMENSIONAL STOCHASTIC, METRIC EXAMPLE.

Figure 10 presents the results of the one-dimensional setup, where the $k$ arms are equally spaced on the range $[0, 1]$. Figure 10(a) shows the regret as a function of time and compares the flat case ($p = 1$) to ABoB with $p = 4$ and $p = 16$ clusters. Figure 10(b) presents the regret at $T = 10k, 50k$, and $100k$ for different numbers of clusters. Both subfigures demonstrate that ABoB has significantly lower regret for the best clustering, $p = 16$, relative to the flat baseline: $1618 \pm 30$ vs. $4866 \pm 54$, respectively (t-test p-value=$9 \times 10^{-25}$), an improvement of 66.8%. Moreover, it shows that ABoB is not much worse (only for $p = 2$, the results were a bit lower due to the symmetry of the problem for $a = \frac{1}{2}$).

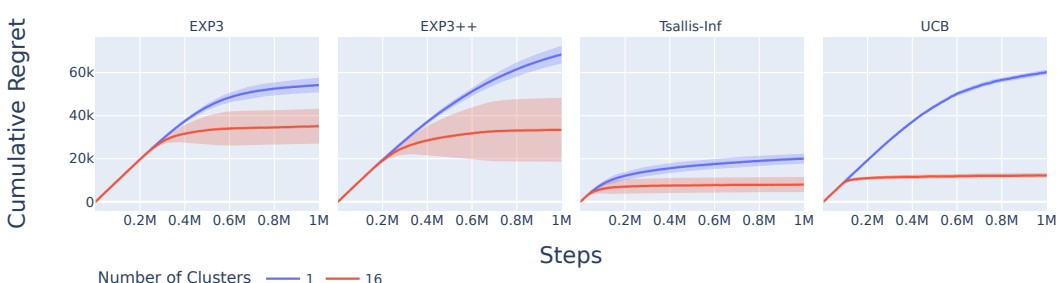

Figure 6: Comparing ABoB (with $\sqrt{k} = 16$ clusters) with "flat" algorithms (single cluster) for different well-known MAB algorithms. Stochastic Scenario from Zimmert & Seldin (2021), $k = 256$.

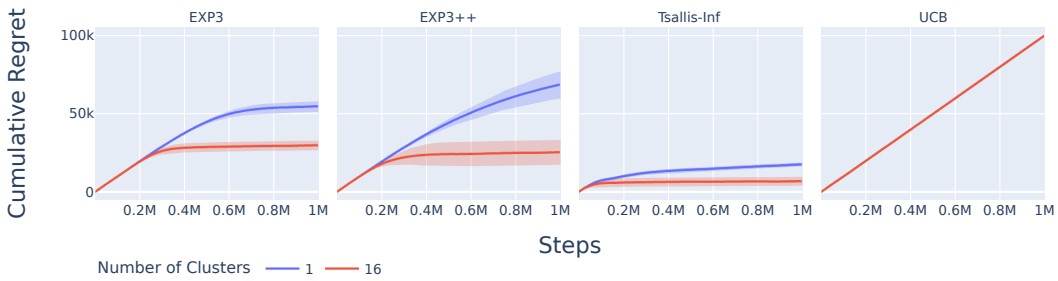

Figure 7: Fixed Optimal Arm, and Nonstochastic Scenario from Zimmert & Seldin (2021), $k = 256$. Comparing ABoB (with $\sqrt{k} = 16$ clusters) with "flat" algorithms (single cluster) for different well-known MAB algorithms.

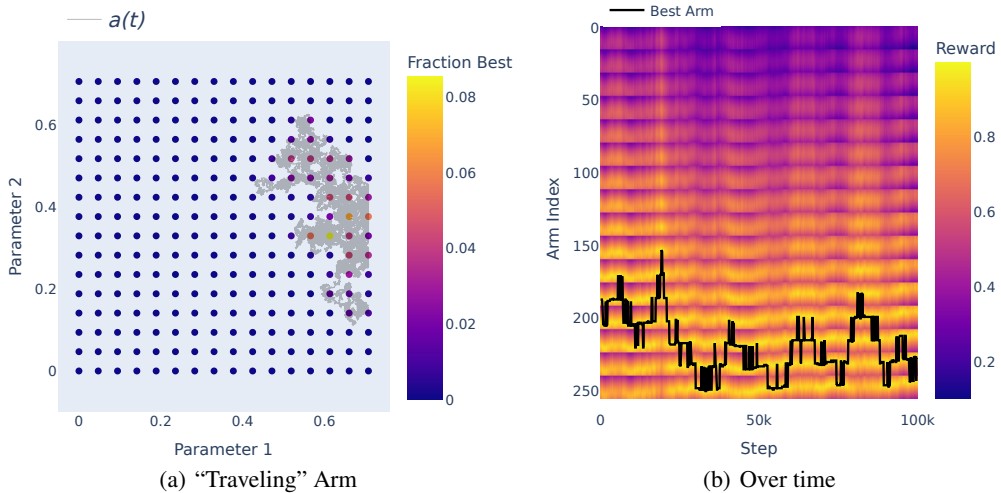

(a) "Traveling" Arm

(b) Over time

Figure 8: The "Traveling" Optimal Arm, Nonstochastic (adversarial) and Metric Space Scenario. (a) Example of the travel of the optimal arm in the configuration space and how often an arm was the best. (b) The arms' mean reward over time and the optimal arm index that changes over time.

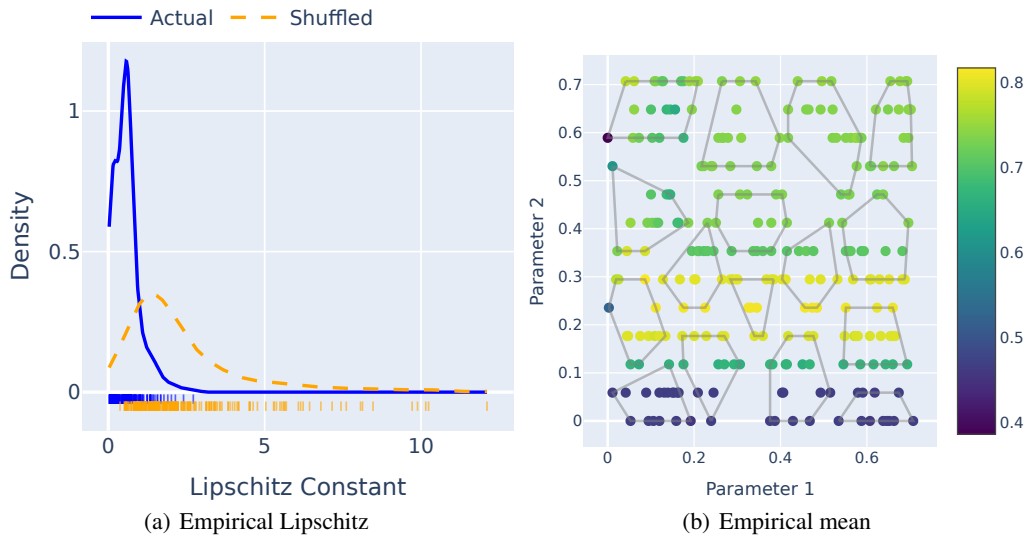

Figure 9: Results from a real storage system. (a) Distribution of the estimation of the Lipschitz constant for the empirical reward function vs. a shuffled one. (b) Empirical mean of each arm on the storage system. The dashed line polygon represents the partition over arms used in ABoB.

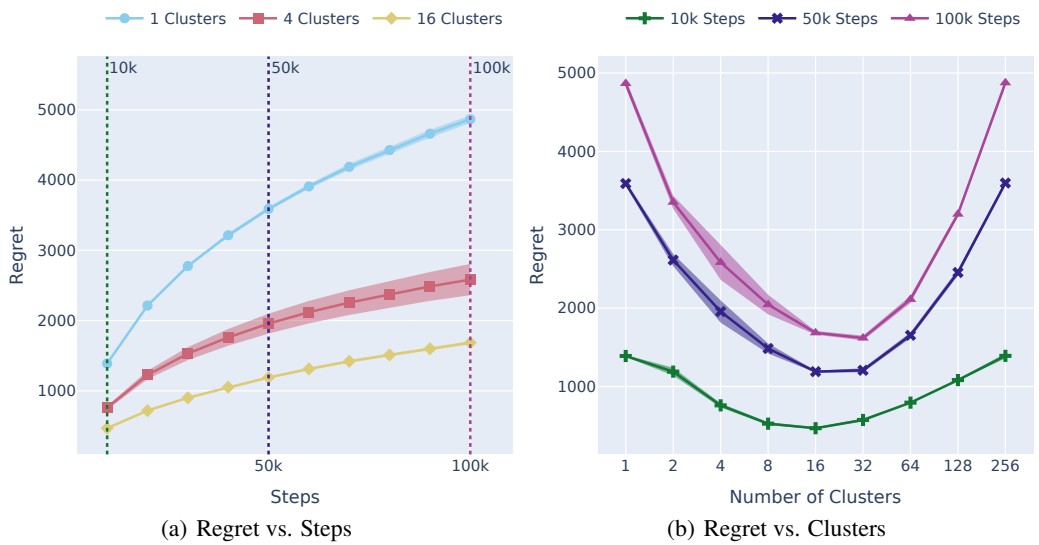

Figure 10: Cumulative regret for ABoB vs. flat Tsallis-INF ($k = 1$ and $k = 256$) for the 1D setup. (a) As a function of steps. (b) As a function of the number of clusters.

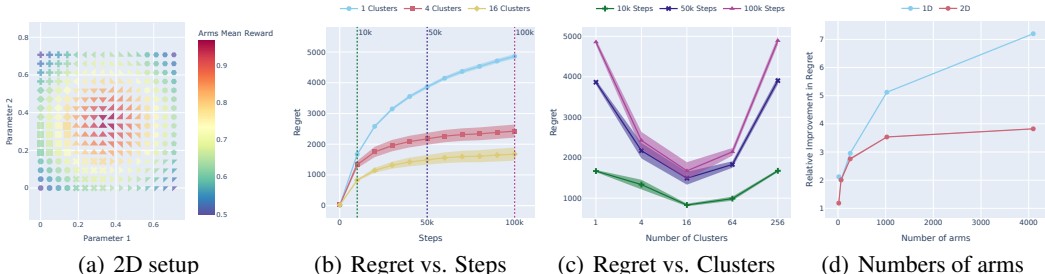

| (a) 2D setup | (b) Regret vs. Steps | (c) Regret vs. Clusters | (d) Numbers of arms |

Figure 11: Cumulative regret for ABoB vs. flat Tsallis-INF ($k = 1$ and $k = 256$) for the 2D setup. (a) A 2D stochastic setup where each marker is an arm with a different reward, and arms are partitioned into clusters. Rewards and the partition are according to Section 5. (b) As a function of steps. (c) As a function of the number of clusters. (d) Effect of the number of arms on the relative regret of ABoB to flat Tsallis-INF for the 1D and 2D cases. More arms imply a larger ratio.

Another important property is the U shape appearing in Figure 10(b). This is a result of Eq. equation 4 from Theorem 4.4 where the first term is monotone increasing with $p$, while the second term is monotone decreasing with $p$, leading to the U shape when the third term is not dominating.

### A.5 Two-Dimensional Stochastic, Metric Example.

In order to verify the extendability to a higher dimension, we repeated the experiment for two dimensions $d = 2$, i.e., we are trying to optimize two parameters. Figure 11(a) depicts the arms' location in the metric space where each marker corresponds to a different cluster, and the color map indicates the expected reward. The Lipschitz condition is held by the setup we use (Section 5). Figure 11 again depicts the benefit of ABoB over using a flat Tsallis-INF, Figure 11(b) as a function of time and Figure 11(c) as a function of the number of clusters. For ABoB the regret at $T = 100k$ was $1675 \pm 213$ and for the flat Tsallis-INF $4859 \pm 72$, respectively (t-test p-value=$7.5 \times 10^{-14}$), an improvement of about 66.5%.

### A.6 Effect of Number of Arms

The previous two sections clearly demonstrated the benefit of ABoB over the flat Tsallis-INF. In order to study the effect of the number of arms $k$ on that benefit we iterated over $k = 2^j$ where $j \in [4, 14]$. Figure 11(d) measures the ratio of the ABoB regret relative to the flat one and shows that the benefit increases as the number of arms increases. This is indeed expected from the results Section 4.2. We note also that, as expected, a larger dimension will require more arms to show the benefit.

### A.7 Effect of Random Clustering

Our setup for the nonstochastic (adversarial) and metric spaces scenario ("traveling arm") in Section 5 considers clustering of the arms by the Lipschitz condition of the 2D metric. Our theoretical results indicates that for any clustering we cannot lose too much. Figure 12 repeat the same experiment of Figure 4 but with random partition of the arms. As we can see, also in this case the clustering improves the regret, but to a smaller extent.

The best result obtained by using 4 clusters, where ABoB achieves a regret of $2086 \pm 491$ compared to the flat Tsallis-INF baseline which achieves a regret of $4166 \pm 291$, (t-test p-value $10^{-8}$). This is about 50% improvment, but less than the 91% improvement achieved by the Lipschitz-based clustering.

### A.8 Bandit Algorithm

In our results, we have considered the use of the Tsallis-INF Multi-Armed Bandit algorithm in our hierarchical approach. Here we show how this approach holds well for other MAB algorithms. Figure 13 shows the regret achieved by using flat vs hierarchical EXP3, which demonstrates that

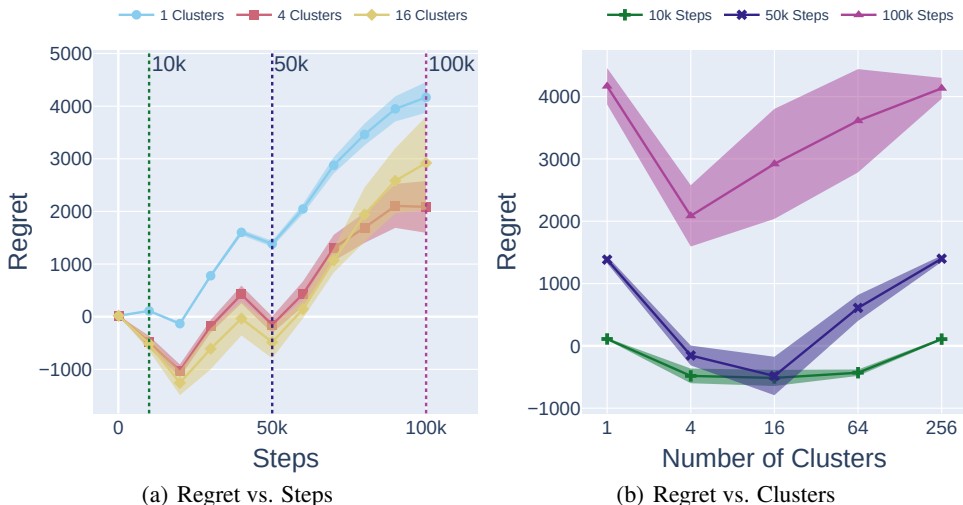

(a) Regret vs. Steps   (b) Regret vs. Clusters

Figure 12: The "Traveling" Optimal Arm, Nonstochastic (adversarial) and Metric Space Scenario with **random clustering**. (a) The cumulative regret for ABoB as a function of steps. (b) The cumulative regret for ABoB as a function of the number of clusters.

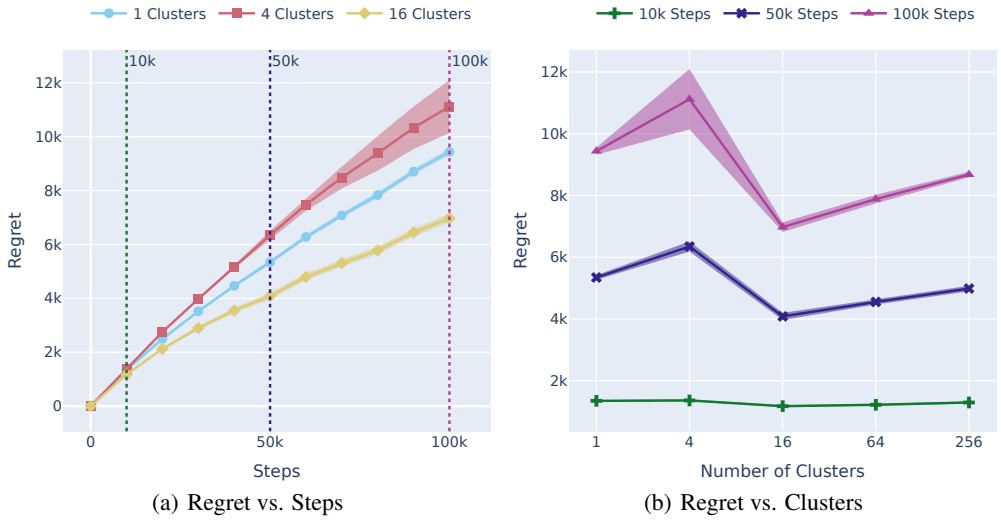

(a) Regret vs. Steps   (b) Regret vs. Clusters

Figure 13: Same as Figure 5 but using EXP3

the hierarchical approach still dominates the flat algorithm (even if overall it performs worse than Tsallis-INF).

## A.9    STOCHASTIC SETUP ON A REAL SYSTEM

In our experimental results (Section 6), we have considered a scenario where the system experiences dynamically changing workloads, which puts us in the non-stochastic setting. Here we show that under a static workload (i.e. stochastic setting) we achieve better performance in regret. Under this scenario, Figure 14 shows that the hierarchical using 16 clusters achieves a regret of $3708 \pm 45$ (compared to $5349 \pm 46$ in the dynamic setting) which is dominant over the flat approach achieving regret of $5587 \pm 26$ (compared to $7568 \pm 43$ in the dynamic setting) (t-test p-value $1.19 \times 10^{-22}$).

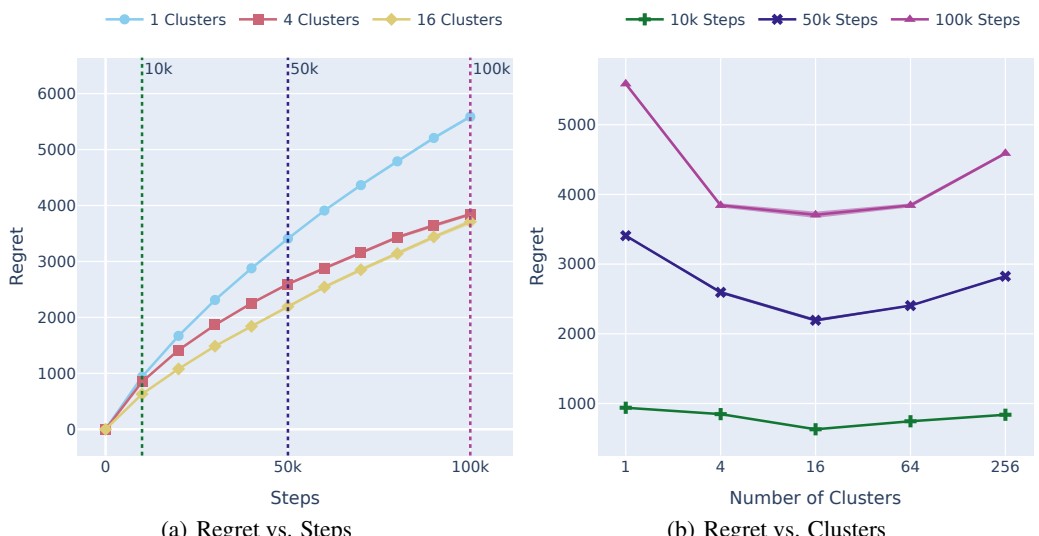

(a) Regret vs. Steps        (b) Regret vs. Clusters

Figure 14: Same as Figure 5 but the system experiences a static workload.

## A.10 DIFFRENT PARENT-CHILD ALGORITHMS

Figures 15 and 16 presented possible combinations of different parent and child algorithms. Note that for the same parent and child algorithm (diagonal), the case of 1 and 256 clusters is the same. For all other cases, in each row, the case of 256 clusters remains the same (only child algorithm works), and for each column, the case of a single cluster remains the same (only parent algorithm works).

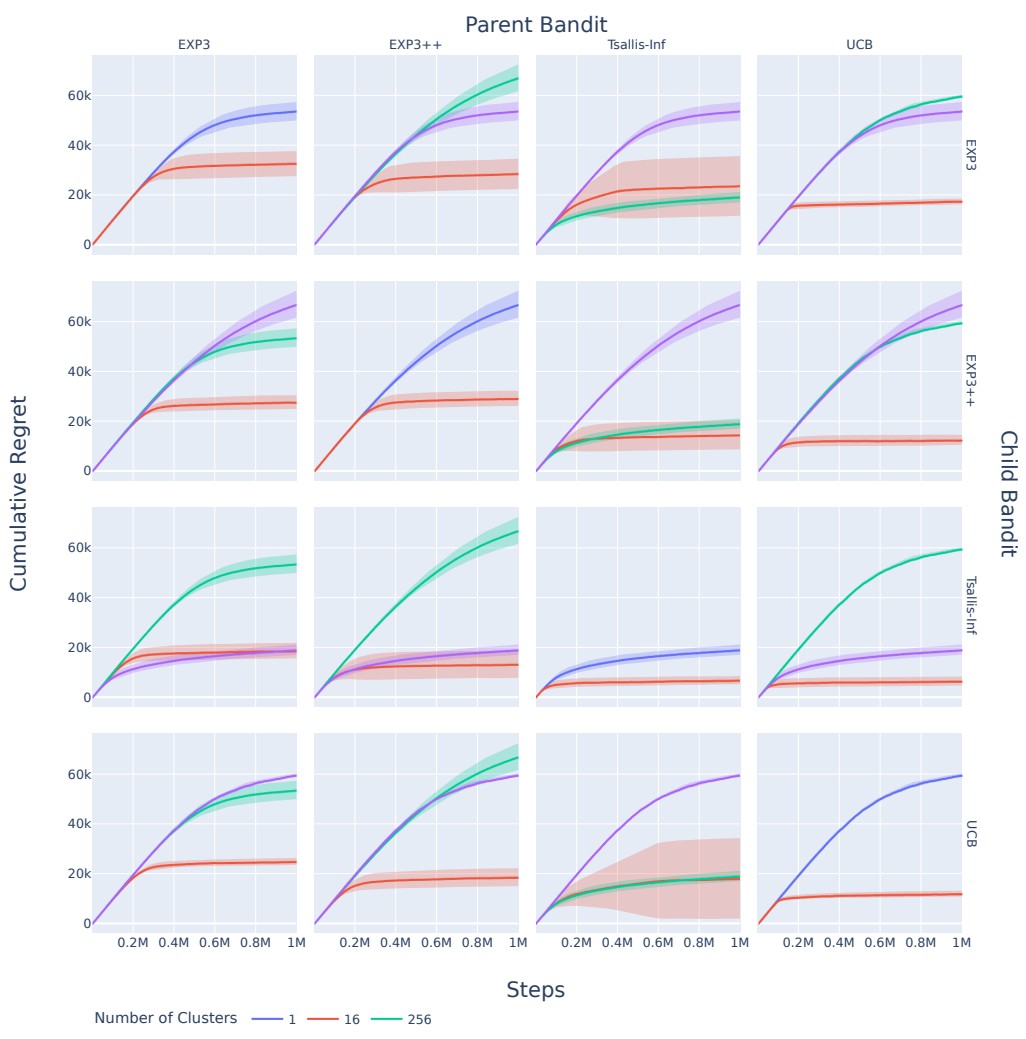

Figure 15: Comparing different parent-child combinations in ABoB. Stochastic environment as in Figure 6

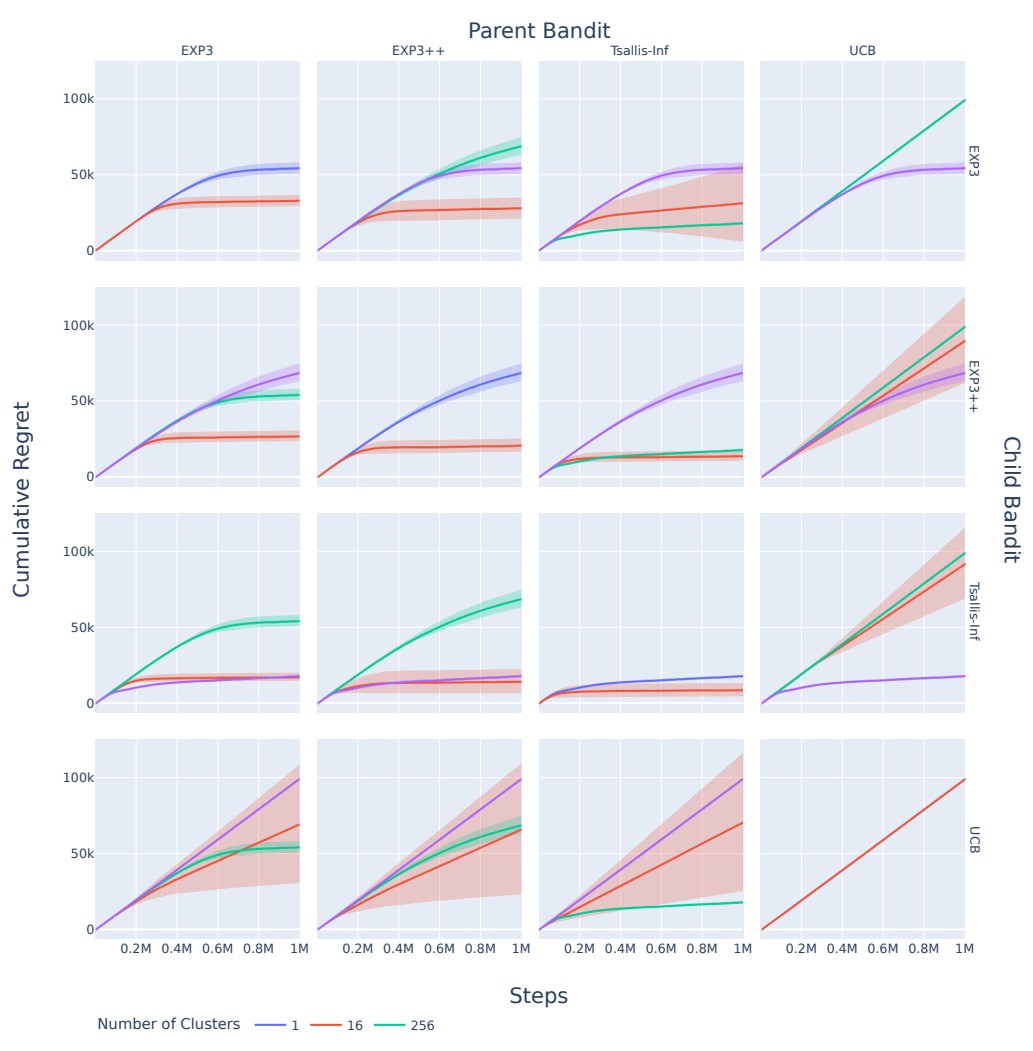

Figure 16: Comparing different parent-child combinations in ABoB. Nonstochastic environment as in Figure 7

