# OpenReview forum: "Hierarchical Bandits for Adversarial Online Configuration Optimization"
_ICLR.cc/2026/Conference — Submitted to ICLR 2026_

### Official Review · Reviewer_L8xs · 2025-10-28

**Soundness:** 2
**Presentation:** 3
**Contribution:** 2
**Rating:** 4
**Confidence:** 3

**Summary:**

The paper proposes ABoB (Adversarial Bandit over Bandits), a hierarchical multi-armed bandit algorithm for adversarial environments with large, structured action spaces. It runs a parent bandit over clusters and a child bandit within each cluster. ABoB matches the worst-case regret of flat algorithms $O(\sqrt{kT})$ and improves to $O(k^{1/4}\sqrt{T})$ under Lipschitz-smooth clusters. Experiments on synthetic data and a real storage system show ABoB outperforms baselines.

**Strengths:**

- The hierarchical architecture is clear and intuitive, providing a simple and modular way to incorporate clustering information without redesigning standard adversarial bandit algorithms.
- The empirical evaluation is comprehensive, covering both synthetic and real-world datasets, with convincing results on a real storage system.

**Weaknesses:**

- Theorem 4.4 and Corollary 4.5 hold only conditionally (under the existence of an idealized adversarial Lipschitz bandit ALB+ that satisfies Property 1). However, the authors later acknowledge that “we are not aware of an algorithm that formally fulfills Property 1 for the Lipschitz condition.” This makes the improved bounds in Theorem 4.4 and Corollary 4.5 hypothetical rather than constructive.
- The improvement in Corollary 4.5 relies on an artificially idealized $l$-partition with $l\leq \frac{1}{4\sqrt{ k }}$ and uniformly sized clusters. This assumption seems too strong.
- Prior studies on hierarchical bandit models, such as [1], are not cited or discussed.


[1] Hong, Joey, et al. "Deep hierarchy in bandits." _International Conference on Machine Learning_. PMLR, 2022.

**Questions:**

See Weaknesses.

---

### Official Review · Reviewer_ZyGH · 2025-10-31

**Soundness:** 3
**Presentation:** 2
**Contribution:** 2
**Rating:** 4
**Confidence:** 3

**Summary:**

This paper introduces a hierarchical algorithm designed to address the nonstochastic Multi-Armed Bandit (MAB) problem in metric action spaces with oblivious Lipschitz adversaries. The setting is motivated by practical challenges of dynamic parameter optimization for systems with finite (albeit large) configuration spaces such as computing and storage systems. Adversarial Bandit over Bandits (ABoB) algorithm structures the large set of configurations by partitioning them into clusters of similar configurations and treating them as virtual arms. The algorithm employs a two leveled hierarchy which includes a parent algorithm (e.g., EXP3 or Tsallis-INF) that selects a cluster, and a child algorithm (also often EXP3 or Tsallis-INF) that selects the physical arm within that cluster. The primary theoretical contribution is showing that this hierarchical approach guarantees a standard worst-case regret $O(\sqrt{kT})$ which matches the traditional MAB approaches. Furthermore, under favorable conditions related to cluster properties and Lipschitz constraints, this paper shows an improved regret bound that can reach $O(k^{1/4}T^{1/2})$. The effectiveness of ABoB is validated through extensive simulations in various stochastic and nonstochastic scenarios and experiments conducted on a real storage system.

**Strengths:**

1. The paper models dynamic parameter optimization in large, structured systems using the nonstochastic MAB framework which is an area with high practical impact.

2. ABoB approach provides a kind of robustness. Regardless of how the arms are clustered, the worst-case regret matches the best known flat adversarial regret, $O(\sqrt{kT})$, which tackles an important barrier to deploying hierarchical methods in dynamic environments.

3. The algorithm demonstrates substantial performance gains across multiple scenarios, especially in the adversarial metric space case. The results on the real storage system further solidify the practical utility of the hierarchical approach.

4. The concept of using parent and child MAB algorithms (like EXP3 or Tsallis-INF) to manage cluster selection and intra-cluster exploration simultaneously is intuitive to incorporating metric structure into adversarial bandits.

**Weaknesses:**

1. The primary theoretical demonstration of the benefit ($O(k^{1/4}T^{1/2})$) relies on the existence of an $ALB^+$ algorithm fulfilling Property 1. This reliance diminishes the supposed theoretical improvement since the actual implementation uses existing MAB algorithms like EXP3 and Tsallis-INF that may not fully utilize the Lipschitz structure in the manner theorized.

2. While Algorithm 1 mentions updating cluster-level parameters using the observed reward $c_t$ (Step 7(e)), the explicit procedure for translating the single observation from the pulled arm $a_t$ into a reward for the virtual arm $P_t$ that is compatible with the parent algorithm (e.g., for EXP3 weights) is not fully detailed. Standard adversarial bandit algorithms require estimating the reward of unpulled arms/clusters as well. Clarification on how the Lipschitz structure is specifically leveraged in this estimation for the parent level would be beneficial.

3. The $\ell$-partition definition in Eq. (3) assumes a maximum reward difference $\ell$ across any two arms in the cluster. While simple, this fixed $\ell$ does not utilize the metric distance $D(a, a')$ specified in the problem formulation in Eq. (1). Leveraging the full distance metric might allow for tighter regret analysis within the cluster, especially if $\ell$ is large but the arms pulled are very close to the optimal arm.

**Questions:**

1.  In the practical implementation of ABoB, i.e., Algorithm 1; how is the expected reward of the selected virtual arm, $c_t(P_t)$, determined to update the parent A-MAB (Step 7(e))? Is this done by simply using the observed reward of the physical arm $c_t(a_t)$? Or, is some form of reward estimation used across $P_t$ to better utilize the cluster's Lipschitz property at the parent level?

2.  Since the improved theoretical result (Theorem 4.4, Corollary 4.5) depends heavily on the existence of the $ALB^+$ algorithm (Property 1), could you provide additional discussions on what properties this "ideal" algorithm must possess? What challenges prevent standard algorithms like Tsallis-INF from formally satisfying Property 1 in the current setting?

3.  In Section 7, the authors suggest future work includes exploring adaptive clustering techniques. Can you comment on how dynamic or adaptive clustering would integrate with the current theoretical framework, especially since Theorem 4.2 relies on a fixed partition $P$? Would dynamic clustering inherently require a penalty term? Would it weaken the "not much to lose" comment in Line 110 and Section 4.1?

---

### Official Review · Reviewer_Cuzn · 2025-11-01

**Soundness:** 1
**Presentation:** 2
**Contribution:** 1
**Rating:** 2
**Confidence:** 4

**Summary:**

The paper proposes a hierarchical “bandit over bandits” scheme (ABoB) for adversarial multi-armed bandits in metric action spaces. A parent adversarial bandit selects a cluster (treated as a virtual arm) and a child bandit inside the chosen cluster selects a concrete arm. For arbitrary clustering, the paper claims a regret on the order of the flat EXP3 baseline via Theorem 4.2; under an intra-cluster Lipschitz condition it introduces Property 1 for a hypothetical “adaptive Lipschitz” child algorithm and derives improved bounds (Theorem 4.4 and corollaries) that scale with the cluster Lipschitz constant $\ell$. Experiments on synthetic data and a storage system show lower regret than flat baselines. The authors emphasize that they have not seen this parent–child structure in prior work and highlight “not much to lose, much to gain” as the main message.

**Strengths:**

- This paper is well-motivated and attempts to solve an important problem of obtaining adaptive guarantee in bandits.
- The designed algorithms are intuitive and empirical results show the effectiveness of the proposed algorithms.

**Weaknesses:**

- One major concern is the missing and mispositioned prior work on bandit-over-bandits. The paper claims novelty for a parent–child bandit scheme and states it has not encountered similar setups. However, closely related “bandit over bandits” frameworks already exist, including corralling/hedging meta-bandit methods (e.g., [Agarwal et al., 2017; Cheung et al., 2021]). These are not cited or compared, while the paper explicitly positions its hierarchy as novel. This leaves the contribution unclear and the related-work positioning incomplete.
- There are also multiple correctness issues in the analysis/theorem statements. Specifically, Theorem 4.2 upper-bounds total regret by adding a parent regret term plus per-cluster EXP3 regrets as if the children received standard bandit feedback. In a hierarchical scheme, child algorithms are triggered only when their cluster is selected; moreover the parent sees bandit feedback that depends on the child’s randomization. This coupling inflates the implicit importance-weighted variance relative to running a single EXP3, exactly the issue handled in [Agarwal et al., 2017] by switching to a log-barrier OMD meta-learner with increasing learning rate. The current proof sketch (Claim 4.1 and Theorem 4.2) does not address this variance amplification and treats each level as if independent EXP3 analyses apply off-the-shelf, which is not generally valid.
- In addition, the assumed “Property 1” does not match known lower bounds. More concretely, the key improved guarantee relies on a child algorithm with regret $O\left(\ell\sqrt{kT\log k}\right)$ for adversarial Lipschitz bandits (Property 1). The paper later acknowledges that it is not aware of an algorithm that actually satisfies Property 1, and in practice uses EXP3 with rescaling inside clusters. Without a concrete, valid child algorithm, Theorem 4.4 is not actionable; in addition, gap-style adaptivity of this kind is **known** to be incompatible with adversarial lower bounds even in plain MAB shown in [Gerchinovitz and Lattimore, 2016]. So the improved bound appears to rest on an algorithmic assumption that is unavailable.

[Agarwal et al., 2017] Corralling a Band of Bandit Algorithms, COLT 2017

[Cheung et al., 2021] Hedging the Drift: Learning to Optimize Under Nonstationarity, Management Science 2021

[Gerchinovitz and Lattimore, 2016] Refined Lower Bounds for Adversarial Bandits, NeurIPS 2016

**Questions:**

- Can the authors compare their framework with [Agarwal et al., 2017] and [Cheung et al., 2021]? Specifically, can the authors explain more on why Theorem 4.2 holds without using a more involved meta algorithm proposed in [Agarwal et al., 2017]?
- Can the authors also explain more on Property 1, especially given the counterexample shown in [Gerchinovitz and Lattimore, 2016]?

**Details Of Ethics Concerns:**

None.

---

### Official Review · Reviewer_H7CL · 2025-11-01

**Soundness:** 3
**Presentation:** 3
**Contribution:** 2
**Rating:** 4
**Confidence:** 3

**Summary:**

The paper considers adversarial MAB with a large, finite, metric action space and a given partition of the arms into “Lipschitz” clusters. It proposes ABoB, a two-level “bandit over bandits”: a parent EXP3 (or Tsallis-INF) chooses a cluster, and a child adversarial bandit inside that cluster chooses the arm. The main theoretical claim is that this hierarchy never worsens the standard  $O(\sqrt{kT})$ adversarial regret of flat EXP3, and under favorable cluster/Lipschitz conditions it can improve the bound to $O(k^{\frac{1}{4}}\sqrt{T})$ when
$p = \sqrt{k}$ and the within-cluster Lipschitz constant is very small.

**Strengths:**

The paper is well-motivated by configuration tuning / storage-system experiments, and the empirical section does show that “cluster first, bandit inside” can help in practice when rewards really are locally smooth.

The paper clearly connects to adversarial Lipschitz/metric bandits and explains when their approach can be preferable to just running a single adversarial algorithm on all arms.

**Weaknesses:**

1. The main bound is almost a direct application of standard EXP3 analysis to two levels plus a telescoping-style decomposition (regret to best cluster + regret inside clusters). There is no fundamentally new adversarial analysis trick here, just a structured way to run existing algorithms. A hierarchical “EXP3 over EXP3” is exactly what one would write down once a partition is given. So the contribution is more organizational than conceptual.

2. To get a better dependence on $k$ the paper assumes picking $p = \sqrt{k}$ clusters of size $\sqrt{k}$ , each cluster satisfies a small Lipschitz constant $l$, and we have an inner algorithm that gets regret $O(l\sqrt{kTlog(k)})$ . But the authors explicitly say they “are not aware of an algorithm that formally fulfills Property 1” and fall back to EXP3 in experiments, which doesn’t realize the theoretical gain. So the main asymptotic improvement is built on an algorithmic assumption the paper doesn’t provide. This weakens the technical contribution of Theorem 4.4 and Corollary 4.5.

**Questions:**

The partition is treated as input, and the regret analysis totally ignores the cost or mistakes of clustering. But for adversarial zooming / adaptive partitioning works, the really hard part is to find a good partition that makes within-cluster Lipschitz constants small. Does this weaken the gain of the algorithm because its assumes there is already good clusters which is the hard part for other method?

---

### Official Review · Reviewer_t1BE · 2025-11-07

**Soundness:** 1
**Presentation:** 3
**Contribution:** 1
**Rating:** 2
**Confidence:** 4

**Summary:**

This paper considers the problem of adversarial multi-armed bandits and proposes a two-level hierarchical algorithm to approach this problem. The algorithm is shown to recover the known $\\sqrt{kT}$ rate for adversarial bandits in the worst case, but is further shown to have better empirical performance in the extensive experiments performed by the authors.

**Strengths:**

The problem is well illustrated and reasonably justified by relevant real-world applications, and the presentation is clear.
The most interesting part of this paper revolves around the experimental results, where the authors clearly show how the proposed algorithm improved upon the performance of existing approaches for nonstochastic multi-armed bandits.

**Weaknesses:**

While the problem is justified and presented as reasonable, further reasoning about the actual results of this work makes it clear that the overall content lacks novelty and contributions.
In particular, the initial presentation of the problem makes it appear like a very general setting with general losses over action sets that correspond to metric spaces.
However, the authors only consider the restriction of this problem under the assumption that there are only finitely many arms, hence leading this problem to turn into a plain adversarial $k$-armed bandit problem.
Therefore, there is no improvement compared to the classical result, simply recovering the well known $\\sqrt{kT}$ regret rate in the general case.
Not only that, but this rate is recovered by implementing a considerably complex bandits-over-bandits algorithm, which is likely to lead to a significant overhead in the computational efficiency of the algorithm.

Even if the authors justify this approach by considering cases where there actually exists a clustering of arms, this assumption comes with prior knowledge of the clustering itself (which in realistic scenarios would actually be unknown) as well as a big catch.
For instance, the authors also suggest considering the case where arms are partitioned into $\\sqrt{k}$ known clusters of $\\sqrt{k}$ arms each, where each loss function satisfies a Lipschitz property within each cluster.
Ideally, it would seem at first that there is a remarkable improvement in the regret guarantee of the proposed algorithm, achieving $k^{1/4}\\sqrt{T}$ regret with a $k^{1/4}$ improvement compared to the standard $k$-armed bandit algorithm that ignores the underlying structure of the loss.
Nonetheless, it needs to be remarked that such a result is only possible conditionally on the existence of an algorithm that can satisfy Property 1 (lines 342-343).
The issue is that such an algorithm is not known to exist, as also remarked by the authors themselves (lines 367-368).
Consequently, this whole initially surprising result, which was also a main selling point of this submission, is essentially unsatisfied by currently known algorithms, and the effective contribution is much more contained than one was led to believe by the somewhat misleading presentation.

**Questions:**

1. The authors mention that they haven’t “encountered similar solutions or setups in existing literature” (line 95). How would the setting considered compare to the long line of work on Lipschitz bandits? They’re possibly more general as they consider continuous actions spaces instead of the finite case as in this paper.
2. The authors propose an algorithm that has a two-level hierarchical structure. This idea of hierarchical algorithms is not novel and has been previously pursued, e.g., by the CORRAL algorithm [1] or algorithmic chaining in nonparametric online learning [2]. It is unclear to me how novel the proposed approach actually compares to them, especially because you cited neither of those works (unless I missed this). Could you comment on this point?

**Minor comments and typos:**
- Line 22: there is an extra period
- Mentioning “adversarial” scenarios in experiments can be odd, since it is usually pretty hard to simulate such environments. It would be more sound to clarify this point by talking about “nonstationary” environments instead, since this is what Zimmert & Seldin (2021) actually design in their instances (which the authors of the current submission take inspiration from)
- At lines 284-286 (Claim 4.1) $p^*$ is used but undefined thus far. It is only defined at line 293. It would be better to move its definition to an earlier point.
- Using $p$ for the *number* of clusters and $p^*$ for the *size* of the cluster containing the best arm is a bit confusing.
- Line 210: “than” instead of “then”
- Line 342: the authors probably meant the more specific eq. (3) instead of eq. (1)
- Line 343: specify that the algorithm in question is ALB${}^*$

**References:**

[1] Alekh Agarwal, Haipeng Luo, Behnam Neyshabur, Robert E. Schapire. *Corralling a Band of Bandit Algorithms*. COLT 2017.

[2] Nicolò Cesa-Bianchi, Pierre Gaillard, Claudio Gentile, Sébastien Gerchinovitz. *Algorithmic Chaining and the Role of Partial Feedback in Online Nonparametric Learning*. COLT 2017.

---

### Meta-Review · Area_Chair_1y5h · 2026-01-04

**Summary:**

The primary contribution of this paper is an approach based on a hierarchical Adversarial Bandit over Bandits. However, the manuscript lacks citations and comparative discussions regarding critical prior work in this area, casting doubt on the novelty and positioning of the research. More importantly, serious concerns were raised regarding the validity of the theoretical results; specifically, it is suspected that Property 1, a necessary assumption for achieving the improved regret upper bound, is never satisfied in practice. Despite these fundamental technical and contextual concerns, the authors did not submit a rebuttal.

**Reviewer Concerns:**

Addressed by Rebuttal: None. The authors did not provide a response or any clarification during the rebuttal period.

Outstanding Concerns: All critical issues remain unaddressed, most notably:

- The missing literature review and comparison with existing hierarchical bandit frameworks.

- The technical validity of Property 1 and its implications for the claimed regret bounds.

- A range of other clarifying questions and minor technical concerns raised by the reviewers.

**Reviewer Scores:**

Given that the authors provided no response to the specific technical and positioning concerns, I predict that the scores would have remained unchanged. The concern that a core property (Property 1) is never satisfied is a potentially fatal flaw for a theoretical paper. Without a rigorous counter-argument or clarification from the authors, reviewers would have no reason to upgrade their confidence or their scores. Consequently, the initial negative assessment remains the only logical conclusion.

---

### Decision · Program_Chairs · 2026-01-26

Reject